# Study on the Automatic Identification of ABX3 Perovskite Crystal Structure Based on the Bond-Valence Vector Sum

**DOI:** 10.3390/ma16010334

**Published:** 2022-12-29

**Authors:** Laisheng Zhang, Zhong Zhuang, Qianfeng Fang, Xianping Wang

**Affiliations:** 1Institute of Material Science and Information Technology, Anhui University, Hefei 230601, China; 2Institute of Solid State Physics, Hefei Institute of Materials Science, Chinese Academy of Sciences, Hefei 230031, China

**Keywords:** space group, crystal system, lattice constant, feature descriptor, the bond-valence vector sum, machine learning

## Abstract

Perovskite materials have a variety of crystal structures, and the properties of crystalline materials are greatly influenced by geometric information such as the space group, crystal system, and lattice constant. It used to be mostly obtained using calculations based on density functional theory (DFT) and experimental data from X-ray diffraction (XRD) curve fitting. These two techniques cannot be utilized to identify materials on a wide scale in businesses since they require expensive equipment and take a lot of time. Machine learning (ML), which is based on big data statistics and nonlinear modeling, has advanced significantly in recent years and is now capable of swiftly and reliably predicting the structures of materials with known chemical ratios based on a few key material-specific factors. A dataset encompassing 1647 perovskite compounds in seven crystal systems was obtained from the Materials Project database for this study, which used the ABX3 perovskite system as its research object. A descriptor called the bond-valence vector sum (BVVS) is presented to describe the intricate geometry of perovskites in addition to information on the usual chemical composition of the elements. Additionally, a model for the automatic identification of perovskite structures was built through a comparison of various ML techniques. It is possible to identify the space group and crystal system using just a small dataset of 10 feature descriptors. The highest accuracy is 0.955 and 0.974, and the highest correlation coefficient (R2) value of the lattice constant can reach 0.887, making this a quick and efficient method for determining the crystal structure.

## 1. Introduction

Perovskite is a naturally occurring mineral that has excellent properties that make it popular in many engineering fields. These include ferroelectric and dielectric materials [1,2,3], catalysis [4], ion conduction [5], thin films [6,7], photovoltaic solar energy conversion cells [8,9,10,11], quantum source devices [12], and nanowire laser gain [13]. The structure of perovskite is frequently shown as ABX3, where A and B are two cations with significantly dissimilar radii. There are a lot of compounds with perovskite structures because many elements in the periodic table can replace the elements in the A and B locations. The B-site cation is typically a transition-metal element with a small radius (such as Cr, Mn, or Sc) and occupies the center of the octahedron. It is coordinated with six X anions. The A-site cation (typically an alkali metal, alkaline earth metal, or rare-earth element) occupies the top corner of the cube and is coordinated with 12 X anions, primarily serving to stabilize the perovskite structure. A BX6 regular octahedron is formed by six X anions and body-centered B-site ions, and the BX6 octahedra are regularly aligned to create a three-dimensional network. The space group and lattice constant of the BX6 octahedron change with the tilt or twist, which alters the crystal’s physical characteristics, such as the electronic energy bands and magnetic order. As a result, creating a model that can precisely and automatically identify the structure of unidentified crystalline compounds is essential for material design.

X-ray scanning is used to detect samples’ diffraction curves, which are then fitted using specialized software to examine the crystal structures. This method demands pricey equipment, and the threshold is high. Additionally, it calls for certain professional knowledge and skills for processing experimental data. Numerous material databases that are well-recognized by the academic community, such as the Open Quantum Materials Database (OQMD) [14], Materials Project (MP), and Inorganic Crystal Structure Database (ICSD) [15], have emerged with the development of materials informatics [16,17], which also provides a richer data resource for studying the methods of crystalline materials. In particular, machine learning (ML) algorithms, which represent artificial intelligence algorithms, continue to advance. Rather than requiring the construction of explicit physical models, these algorithms automatically model the linear and nonlinear relationships between these physical variables through probabilistic statistical learning to achieve quick and affordable classification predictions, which have significant implications for the identification and screening of materials on a large scale. Numerous studies on the identification of crystal structures based on deep learning (DL) techniques of XRD patterns have been published recently [18,19,20,21]. These studies have led to significant advances in classifying crystalline materials. However, the identification of multiple crystal structures, in particular, 230 crystal space groups, calls for a substantial amount of XRD data and is sensitive to poor X-ray diffraction data, which do not apply to the data identification of tiny samples. Small sample data can be recognized using machine learning. Traditional ML approaches mainly rely on manually chosen descriptors, which should have a distinct physical meaning. The most frequently used descriptor in the study of materials informatics is the elemental information of the material composition [22,23,24]. Even though efforts have been made to incorporate the ionic radius calculation tolerance factor (t) into the feature set [25,26,27], elemental information based only on the chemical composition does not apply to all perovskite structures, especially to those that have the same composition but differ in structure. To get around the problem brought on by the structural diversity of perovskites, better physical descriptors must be utilized to explain the complicated geometry of these materials.

This work establishes a new perovskite feature set, provides a thorough analysis of the variables used to characterize ABX3-type perovskite crystals, introduces the bond-valence vector sum (BVVS) descriptor with a clear physical meaning to capture the intricate geometry of perovskite, and creates an intelligent, affordable, and reliable model to identify unidentified crystalline compounds with a small dataset of only 10 feature descriptors. The crystal system and space group that the crystals belong to can be determined with accuracy from a small dataset of only 10 feature descriptors, and the lattice constants can also be predicted with accuracy.

## 2. Materials and Methods

### 2.1. Data Acquisition

The Materials Project, a well-known materials science database, and the related literature were the sources of all of the data used in this study. From the database, we pulled 1647 records with stable perovskite structures, spanning 40 space groups and 7 crystal systems. The distribution of the gathered lattice constants a, b, and c ranges from 2 Å to 11 Å. The stability of the perovskite structure must be taken into account when gathering data, and the Goldschmidt tolerance factor *t* [28] can be used in calculations to determine whether the perovskite structure can be created. Its equation is as follows:(1)$$t=\frac{r_{A}+r_{B}}{\sqrt{2}\left(r_{B}+r_{X}\right)}$$
where *r_A_*, *r_B_*, and *r_X_* are the effective ionic radii of the *A*-site, *B*-site, and *X*-site, respectively, and the value of *t* is equal to 1 in an ideal cubic perovskite structure. Generally, perovskite can be formed in the 0.8 < *t* < 1.0 range.

### 2.2. Feature Engineering

In ML, feature engineering is a crucial stage. The construction, extraction, and selection of features are all parts of feature engineering. Among these, feature selection primarily serves to prevent the model from overfitting and enhance the model’s capacity for generalization. Feature descriptors are a crucial component of the ML approach. The descriptor set of the model can theoretically include any feature descriptor that can reflect the crystal structure, but redundant feature descriptors will hurt the final model’s accuracy and computational efficiency. Investigating the key feature factors that most influence the goal features is essential. This work focused on screening the key structural feature descriptors with physical significance after extracting as many potential nonlinear relationship features between atomic parameters and crystal structure from the database Materials Project as we could.

Significant structural variations are caused by the unique atomic characteristics of the perovskite’s constituent elements. This is because of the abundance of voids, which are prone to lattice distortion, between the BX6 octahedra. The BX6 octahedron is susceptible to skew rotation and defects when the ionic radii of the A and B sites are too dissimilar. To quantify the BX6 octahedral distortion and explain its physical characteristics in terms of intracrystalline chemical bonding, we present the modulus of the bond-valence vector sum (BVVS). Bond-valency theory [29] states that each atom wants a bond-valency sum equal to its atomic valency; however, the actual atomic valency can be determined by adding the bond valencies of the bonds that connect that atom to its neighbors. Here, the relationship between the bond valence and bond length can be expressed by the following equation.
(2)$$S_{ij}=\exp\left(\frac{R_{0}-R_{ij}}{b}\right)$$
where *b* is a constant of 0.37 Å, *R*_0_ is an empirical constant related to the type of atom (ion), $S_{ij}$ is the bond valence between atom *i* and atom *j*, and $R_{ij}$ is the bond length between atom i and atom j and can be determined from the inorganic crystal structure database. Since the bond valence $S_{ij}$ is directional, to take this directional feature into account, the bond valence vector ${\vec{S}}_{ij}$ can be defined as:(3)$${\vec{S}}_{ij}=S_{ij}{\vec{R}}_{ij}$$
where ${\vec{R}}_{ij}$ is the unit vector from atom *i* to atom *j*. I. D. Brown [30] proposed a bond-valence sum rule based on the electrovalence rule: i.e., the bond-valence sum of the chemical bonds attached to each atom is equal to the valence state of that atom. By summing the ${\vec{S}}_{ij}$ values, the atomic valence is obtained, expressed as the BVVS, and can be calculated by the following equation.
(4)$${\vec{V}}_{i}=\underset{i\neq j}{\sum }{\vec{S}}_{ij}$$
where ${\vec{V}}_{i}$ is the atomic valence state, and ${\vec{V}}_{i}$ is zero in the stable coordination sphere and is not zero when distortion occurs. Figure 1 shows a schematic diagram of the BVVS. The center is a B atom; ideally, the BVVS is zero, and when BX6 octahedral distortion occurs, the BVVS is non-zero.

By assigning the valence state to the chemical bonds arranged around the atoms, the BVVS can link the valence state to the crystal structure, making it possible to study the crystal structure using the valence of the chemical bonds. Therefore, we added the modulus of the BVVS to the set of constituent element features, so the original set of 24 feature descriptors based on the constituent element features and structural features is created, as shown in Table 1.

There are many parameters used to describe atomic information, e.g., atom radius, but these feature descriptors do not play an equal role in the construction of the crystal structure. In other words, some descriptors have a stronger relationship with the crystal structure than others, which can accelerate the convergence in the right direction more easily and reduce the computational effort of model training. Support vector machine regression (SVR) was employed by Takahashi et al. [31] to predict the lattice constants of 1541 binary body-centered cubic crystals, with an R2 value of 0.836. The characteristic descriptors used included atomic number, atomic radius, electronegativity, electron affinity, atomic orbital, and valence electron number. Jarin et al. [32] predicted the type of crystal structure and its lattice parameters using the basic atomic properties of perovskite materials. Atomic number, atomic mass, valence, ionic radius, electronegativity, and the polarizability of A and B atoms are some examples of these atomic attribute signals. They found that atomic characteristics such as ionic radius, electronegativity, bond-valence vector, atomic radius, number of atoms, and covalent radius strongly correlate with the crystal structure. Based on their research, we chose some widely accepted atomic parameters as initial descriptors and used the recursive feature descriptor method to remove irrelevant and weakly correlated atomic parameters while keeping the same model accuracy constant. Finally, we chose the retained features shown in Table 2. In this way, we constructed a 1647-perovskite dataset with a total of 10 features of perovskite constituent element features and structural features. The mean values and standard deviations of all features’ A, B, and X positions were calculated as inputs to ensure that each compound can acquire the same number of features and properly understand the data features. In the meantime, some empty data were removed, and 90% of the training set and 10% of the test set were partitioned at random.

### 2.3. Machine Learning Modeling

ML algorithms come in two flavors: classification and regression. Regression and classification models were both extensively used in this work. ML algorithms were compared, and the optimal algorithm model was ultimately chosen. These include the widely used Support Vector Machines (SVC), Extreme Gradient Boosting (XGBoost), Gradient Boosting Trees (GBDT), and Random Forest (RF).

### 2.4. Model Evaluation

The mean absolute error (MAE), mean square error (MSE), and correlation coefficient (R2) in the ML regression model are primarily used to assess the prediction accuracy of the material system model. Better model performance and greater prediction accuracy are shown by smaller MAE and MSE and larger R2. These are the equivalent equations:(5)$$\mathrm{MSE}=\frac{1}{n}\overset{n}{\underset{j=1}{\sum }}{\left({\hat{y}}_{j}-y_{j}\right)}^{2}$$
(6)$$R2=1-\frac{\sum_{j=0}^{n-1}{\left({\hat{y}}_{j}-y_{j}\right)}^{2}}{\sum_{j=0}^{n-1}{\left(y_{j}-{\bar{y}}_{j}\right)}^{2}}$$
(7)$$\mathrm{MAE}=\frac{1}{n}\overset{n}{\underset{j=1}{\sum }}\left|{\hat{y}}_{j}-y_{j}\right|$$
where *n* denotes the number of samples, $y_{j}$ is the true value, ${\hat{y}}_{j}$ is the predicted value, and ${\bar{y}}_{j}$ is the mean value. The accuracy of the classification model is mainly evaluated by accuracy (ACC), the Matthews correlation coefficient (MCC), and the balanced F-score (F1-score). The larger the ACC, the higher the accuracy of the prediction; the larger the MCC, the higher the correlation between the prediction and the actual result; and the larger the F1-score, which takes into account the calculation of the accuracy and completeness of the model, the higher the quality of the model.

## 3. Results

### 3.1. ML Algorithm Analysis

On the feature set without the BVVS, we first pre-trained several ML algorithm models (all with default parameters), and we then compared how well each model identified perovskite crystal systems and space groups to choose the best model. For the 1647-perovskite dataset, we divided the training set into 90% and the test set into 10% at random. The models with superior effects when recognizing 7 crystal systems and 40 space groups are RF, XGBoost, and GBDT, whereas the worst model is SVC. Table 3 displays the classification results for seven crystal systems on the SVC, RF, GBDT, and XGBoost test sets, while Table 4 displays the classification results for 40 spatial groups on the four ML test sets. Among them, RF has the highest accuracy (ACC), Matthews correlation coefficient (MCC), and balanced F-score (F1-score) in identifying crystal systems and space groups, but SVC is the worst. Altogether, RF has the best performance, so all of the next experiments were performed using RF.

### 3.2. BVVS Analysis

We conducted two sets of comparative tests before and after adding the BVVS to investigate the significance of the BVVS feature descriptor. Before and following the addition of the BVVS, respectively, the crystal system and perovskite space group were determined using RF, and the lattice constants were predicted. Table 5 contains the final RF hyperparameter settings. Figure 2a displays the test set identification results for seven crystal systems using the RF classification technique. The vertical coordinates correspond to the relevant particular values, while the horizontal coordinates represent the model performance metrics. ACC rose from 0.915 to 0.974, MCC increased from 0.883 to 0.961, and the F1-score increased to 0.970 after the addition of the BVVS. The results of the RF test set identification for 40 space groups of gathered perovskite are shown in Figure 2b. The 40 space groups’ identification accuracy (ACC), Matthews correlation coefficient (MCC), and equilibrium F-score (F1-score) had values of 0.806, 0.756, and 0.796, respectively, before the addition of the BVVS. After the addition of the BVVS, the ACC increased to 0.955, the MCC had a value of 0.943, and the F1-score increased to 0.947. The BVVS is crucial in determining the crystal shape. With the inclusion of the BVVS, the identification of crystal systems and space groups is greatly improved. Figure 2c shows the fitting results of the predicted lattice constant a on the test set of the RF regression model before adding the BVVS, where the horizontal coordinate is the true value of the lattice constant, the vertical coordinate is the predicted value, and the highest correlation coefficient R2 of the prediction is only 0.710. The results of the projected lattice constants fitted on the test set after the addition of the BVVS are shown in Figure 2d. The greatest R2 after the addition of the BVVS reaches 0.887, the MAE and MSE have also been greatly reduced, and the overall fitting impact has been significantly enhanced. The aforementioned comparison trials show conclusively that the addition of the BVVS can more correctly reflect the crystal’s structural properties. This is primarily because erections between the atoms that make up the crystal determine its structure and properties. These interactions are reflected in the chemical bonds that connect the atoms, and the behavior of these chemical bonds and associated crystal parameters are crucial characterization variables of such interactions that can be used to distinguish the structural differences between various crystals.

## 4. Discussion

In contrast to earlier research [33,34], we estimated the lattice constants and automatically identified the space group of several perovskite materials using only a small dataset of 10 characteristics. The technique we employed is a combination of physically meaningful feature variables (BVVS) that quantifies lattice distortions relative to the constituent atomic features. This enables ML predictions to be physically interpreted and to be more controllable in the direction of the target feature variable. The accuracy of the crystal system and space group identification is far superior, especially the space group accuracy of 0.955, which is more outstanding, and the lattice constants can also be predicted, as shown in Table 6, when compared with the XRD-based ML method and the feature-descriptor-based methods of other works [18,33,34]. The confusion matrix of our RF recognition method for the crystal system and space group on the test set is shown in Figure 3. The horizontal coordinates are the recognized categories, the vertical coordinates are the true categories, the values in the squares indicate the percentage of the number of row label categories predicted as column label categories, and the larger values and darker color of the diagonal squares represent higher recognition accuracy, whereas the remaining squares with light colors represent lower recognition accuracy. Although the overall level of accuracy for each category identification remains high, certain lower values are directly tied to the sample distribution. Figure 4 depicts the prediction of all lattice constants, including a, b, c, α, β, and γ. When predicting a, b, and c, good accuracy is attained, and the maximum R2 value is 0.887; nevertheless, there is substantial dispersion when predicting angles, which is also probably due to the uneven sample of original data angles and inadequate model learning.

The significance of the BVVS feature descriptors under RF was further assessed, and the ML model feature ranking approach was used to rate the significance of these 10 feature descriptors. The feature variable importance histogram is shown in Figure 5. The ordinate in Figure 5 represents the 10 feature variables, and the abscissa is the feature importance coefficient. The larger the importance coefficient, the greater the contribution to the predicted value of the target feature variable. It is clear from the feature importance histogram that the BVVS makes the largest contribution to the identification of the crystal structure, further demonstrating its ability to effectively capture crystal structure data and support crystal structure identification. It is important to note that, as the bar chart illustrates, the number of atoms also makes a greater contribution to the target characteristic variables. This is because one fundamental characteristic of a crystal cell is the number of atoms present. The more atoms present in a crystal, the more permutations between those atoms, the more resulting distortions, and the more complex the crystal structure. The complexity of the crystal structure and the atom count are closely correlated. Different characteristic factors have varying degrees of influence on the crystal structure, including molar volume, Pauling electronegativity, atomic radius, average ionic radius, covalent radius, etc.

## 5. Conclusions

In conclusion, we provide a novel approach for predicting the crystal structure of perovskite. The atomic characteristic information of the ABX3 perovskite composition is examined, and a new characteristic variable, BVVS, is added. This new characteristic variable is a physically significant combinatorial structural characteristic variable that reflects the outcome of the integrated interaction between various atoms, which can reflect the BX6 octahedral distortion from the perspective of chemical bonding and is a characteristic descriptor that cannot be neglected for quantitatively capturing various complex crystal structures. With the highest identification accuracy of 0.974 and 0.955 for the crystal system and space group and the highest prediction R2 of 0.887 for the lattice constant, we have discovered that RF works best when aggregated across many ML models. Our contribution is that the newly introduced BVVS enables ML to have a physical interpretation, learn precisely in the direction of the target feature variables, and adapt well to small-sample-dataset prediction without building a large dataset. Furthermore, only 10 feature descriptors are required to identify the structure of a crystal, significantly reducing the difficulty of crystal structure prediction. In the meantime, the set of feature descriptors developed in this study may be successfully used to predict the structure of a larger variety of perovskite materials, which also serves as a foundation for predicting a larger number of perovskite-material-related attributes. Additionally, by avoiding costly DFT calculations, the amount of calculation is decreased, making our technique reasonably affordable to utilize. To determine the correlation between the features employed and the predicted crystal structure, we also conducted a feature variable importance analysis. This analysis offers fresh perspectives on how to identify the desired crystal structure for perovskite materials that will be designed in the future. With the growing database of research materials and the development of machine learning algorithms, there are discoveries in the optimization and iteration of these methods, which provide better and faster aid to studies, even though the ML algorithm model used to identify the space group of perovskite materials and predict the lattice constants still has some shortcomings.

## Figures and Tables

**Figure 1 materials-16-00334-f001:**
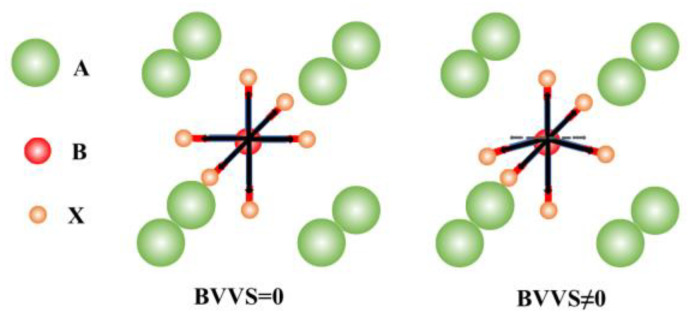
Schematic diagram of the bond-valence vector sum.

**Figure 2 materials-16-00334-f002:**
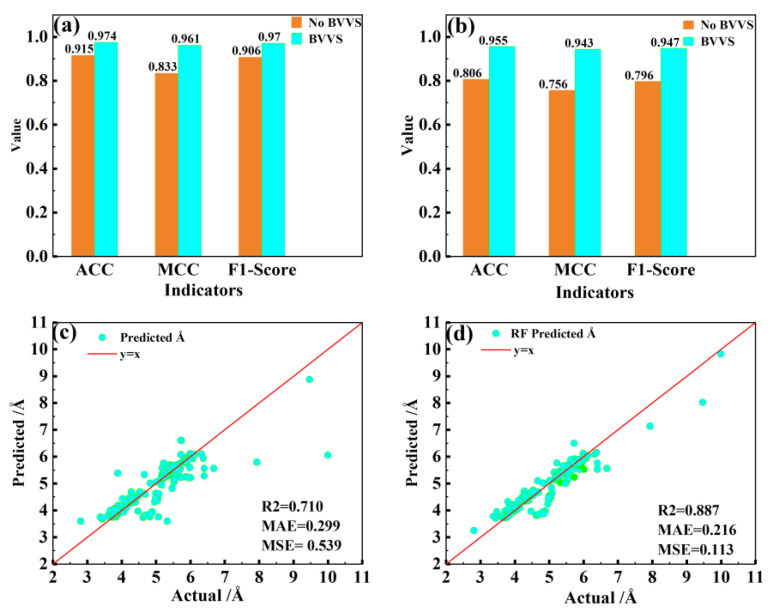
Experimental results before and after adding BVVS: (**a**) crystal system; (**b**) space group; (**c**) the lattice constant a before adding BVVS; (**d**) the lattice constant a after adding BVVS.

**Figure 3 materials-16-00334-f003:**
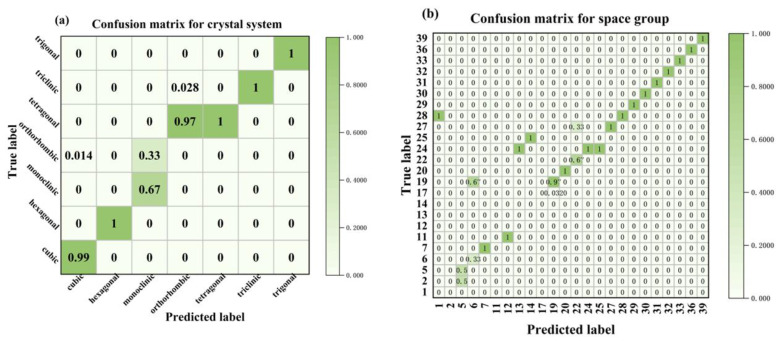
Confusion matrix for RF-identified crystal systems and space groups: (**a**) crystal system; (**b**) space.

**Figure 4 materials-16-00334-f004:**
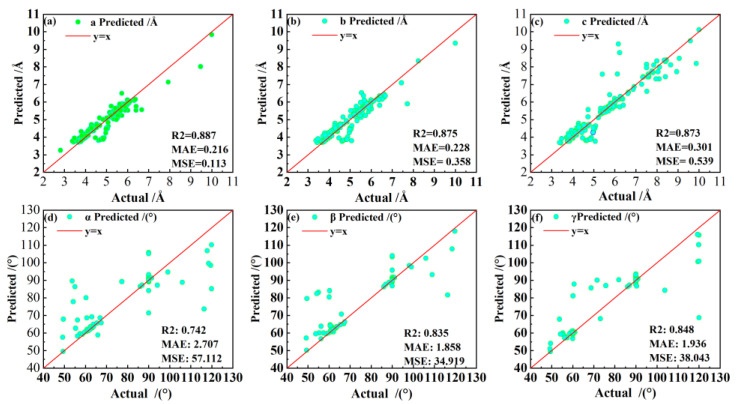
Predicted lattice constants of perovskite: (**a**) a; (**b**) b; (**c**) c; (**d**) α; (**e**) β; (**f**) γ.

**Figure 5 materials-16-00334-f005:**
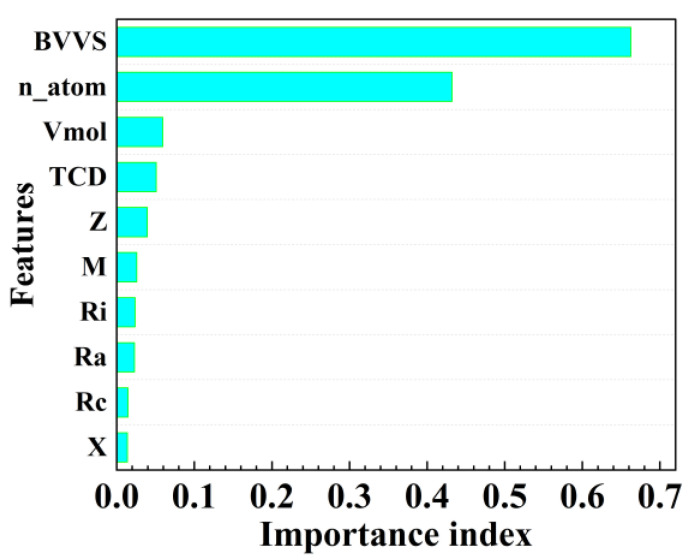
Importance histogram of characteristic variables.

**Table 1 materials-16-00334-t001:** Perovskite original feature descriptor set and its physical meaning.

Descriptors	Physical Meaning	Descriptors	Physical Meaning
n_atom	Number of atoms	TCD	Thermal conductivity
Z	Atomic number	Tb	Boiling point
G	Group in periodic table	Tm	Melting point
P	Period in periodic table	Tc	Critical temperature
M	Atomic mass	Ef	Enthalpy of fusion
Vmol	Molar volume	FIE	First ionization
Ra	Atomic radius	es	The number of electrons in s orbitals
Ri	Average ionic radius	ep	The number of electrons in p orbitals
Rvdw	Van der Waals	ed	The number of electrons in d orbitals
Rc	Covalent radius	ef	The number of electrons in f orbitals
X	Pauling electronegativity	ER	Electrical resistivity
EA	Electron affinity	BVVS	The bond-valence vector sum

**Table 2 materials-16-00334-t002:** Descriptor set after feature selection and their physical meaning.

Descriptors	Physical Meaning
n_atom	Number of atoms
BVVS	The bond-valence vector sum
Z	Atomic number
Ri	Average ionic radius
Ra	Atomic radius
M	Atomic mass
X	Pauling electronegativity
Vmol	Molar volume
Rc	Covalent radius
TCD	Thermal conductivity

**Table 3 materials-16-00334-t003:** Classification results of crystal systems on the four ML test sets.

Algorithm	ACC	MCC	F1-Score
SVC	0.372	0.023	0.207
GBDT	0.898	0.872	0.900
RF	0.915	0.883	0.906
XGBoost	0.853	0.795	0.814

**Table 4 materials-16-00334-t004:** Classification results of spatial groups on the four ML test sets.

Algorithm	ACC	MCC	F1-Score
SVC	0.367	0.048	0.197
GBDT	0.777	0.717	0.767
RF	0.806	0.756	0.796
XGBoost	0.690	0.600	0.626

**Table 5 materials-16-00334-t005:** Final RF hyperparameters.

Hyperparameters	Value
criterion	entropy
n_estimators	100
max_depth	10
n_job	−1

**Table 6 materials-16-00334-t006:** Comparison of crystal structure recognition accuracy.

—	Ours	Park et al. [18]	Liang et al. [33]	Li et al. [34]
Crystal system	0.974	0.949	0.907	0.816
Space group	0.955	0.811	0.638	0.729

## Data Availability

Not applicable.

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
