# Peer review of "Study on the Automatic Identification of ABX3 Perovskite Crystal Structure Based on the Bond-Valence Vector Sum"

_materials, 2022, doi:10.3390/ma16010334_

Round 1
Author Response
Dear Editors and Reviewers:
Thank you for your letter and for the reviewers’ comments concerning our manuscript entitled “Study on the automatic identification of ABX3 perovskite crystal structure based on the bond-valence vector sum”(ID: materials-2124174). Those comments are all valuable and very helpful for revising and improving our paper, as well as the important guiding significance to our research. We have studied the comments carefully and have made corrections which we hope meet with approval. Revised portions are marked in red on the paper. The main corrections in the paper and the responses to the reviewer’s comments are as flowing:
Responds to the reviewer’s comments:
Point 1: In the introduction, lines 31 and 32, it is talked about the applications of perovskites. However, I miss some important applications like for quantum sources devices or as gain material for nanowires lasers, as it was evidenced in references [1-2].
[1] Metal, dielectric and hybrid nanoantennas for enhancing the emission of single quantum dots: A comparative study. Journal of Quantitative Spectroscopy and Radiative Transfer Volume 276, 107900 (2021).
[2] Applications of Hybrid Metal-Dielectric Nanostructures: State of the Art. Advanced Photonics Research. Vol. 3, No. 4 (2022).
Response 1: Thank you for your reminder, we have taken your suggestion and will add references [12],[13] in line 33 of the revised manuscript。
- Barreda, A.; Hell, S.; Weissflog, M.A.; Minovich, A.; Pertsch, T.; Staude, I. Metal, dielectric and hybrid nanoan-tennas for enhancing the emission of single quantum dots: A comparative study. Journal of Quantitative Spec-troscopy and Radiative Transfer 2021, 276, 107900.
- Barreda, Á.; Vitale, F.; Minovich, A.E.; Ronning, C.; Staude, I. Applications of Hybrid Metal-Dielectric Nanostructures: State of the Art. Advanced Photonics Research 2022, 3, 2100286.
Point 2: Could you explain how and why did you choose the selected characteristic descriptors?
Response 2: Thank you for your question about feature descriptor selection. There are many parameters used to describe atomic information, i.e. atom radius, but these feature descriptors don’t play an equal role in the construction of the crystal structure. In another word, some descriptors have a stronger relationship with crystal structure than others, which can accelerate the convergence in the right direction more easily and reduce the computational effort of model training. Firstly, we chose some widely accepted atomic parameters as initial descriptors mentioned in reference [31]、[32]( In lines 162 and 165 of the revised version). Next, we used the recursive feature descriptor method to remove the irrelevant and weakly correlated atomic parameters while keeping the same model accuracy constant. Considering the limitations of atomic feature information alone to describe the crystal structure, we need to find more effective special descriptors. Finally, we experimentally found the bond-valence vector sum (BVVS), a vital feature descriptor that can capture the complex crystal structure of perovskite well. Thus, a total of 10 effective feature descriptors were used to construct the crystal structure with BVVS.
- Takahashi, K.; Takahashi, L.; Baran, J.D.; Tanaka, Y. Descriptors for predicting the lattice constant of body cen-tered cubic crystal. The Journal of Chemical Physics 2017, 146, 204104.
- Jarin, S.; Yuan, Y.; Zhang, M.; Hu, M.; Rana, M.; Wang, S.; Knibbe, R. Predicting the Crystal Structure and Lattice Parameters of the Perovskite Materials via Different Machine Learning Models Based on Basic Atom Properties. Crystals 2022, 12, 157.
Point 3: I miss an explanation of the different cited models: Accuracy (ACC), Matthews correlation coefficient (MCC), and balanced F Score (F1-Score).
Response 3: These three parameters are commonly used in the evaluation metrics for machine learning classification models. Accuracy (ACC) indicates the ratio of the number of correctly classified test instances to the total number of test instances. Matthews correlation coefficient (MCC) can view the prediction and the true outcome as two 0-1 distributions, and then measure the similarity of the two distributions using the Matthews correlation coefficient. F1-Score is based on the summed average of Recall and Precision, i.e., the recall and precision are evaluated together. More detailed information could be found in my uploaded attachments “Classification Model Evaluation Metrics” of (1), (2), (3), (4), (5).
Point 4: I recommend to include a more extended explanation about the importance coefficient of the different feature variables.
Response 4: Thank you for your suggestions. The importance coefficient is used to reveal the relationship between input and output in a prediction model. Feature importance scores can be calculated for problems involving predicted values (called regression) and category labels (called classification). The bigger the importance coefficient value, the greater contribution to the prediction model. We added the interpretation of the contribution of other feature descriptors to the predicted target feature variables in lines 352 to 359 of the revised manuscript.

Reviewer 2 Report
The current manuscript has a detailed description, and each method has been discussed in detail. The English level was acceptable and understandable to the reader.
To begin with the introduction, I should conclude that it was engrossing. A suitable history, in addition to a challenging issue, has been mentioned.
Although the number of paragraphs was less, the description and details were not incomplete.
However, it would be better to explain why the authors decided on machine learning instead of empirical experiments.
For other manuscripts parts, the paragraphs have been designed properly. However, it was a little difficult to understand for the new researchers who desired to construct their knowledge according to this manuscript.
Author Response
Dear Editors and Reviewers:
Thank you for your letter and for the reviewers’ comments concerning our manuscript entitled “Study on the automatic identification of ABX3 perovskite crystal structure based on the bond-valence vector sum”(ID: materials-2124174). Those comments are all valuable and very helpful for revising and improving our paper, as well as the important guiding significance to our research. We have studied the comments carefully and have made corrections which we hope meet with approval. Revised portions are marked in red on the paper. The main corrections in the paper and the responses to the reviewer’s comments are as flowing:
Responds to the reviewer’s comments:
Point 1: The current manuscript has a detailed description, and each method has been discussed in detail. The English level was acceptable and understandable to the reader. To begin with the introduction, I should conclude that it was engrossing. A suitable history, in addition to a challenging issue, has been mentioned. Although the number of paragraphs was less, the description and details were not incomplete. However, it would be better to explain why the authors decided on machine learning instead of empirical experiments. For other manuscripts parts, the paragraphs have been designed properly. However, it was a little difficult to understand for the new researchers who desired to construct their knowledge according to this manuscript.
Response 1: Thank you for your recognition and questions. Experimental validation is the only criterion to judge whether the theory is correct. Instead of traditional empirical experiments, we decided to take a try by using machine learning method to explore the feasibility of crystal structure prediction based on simple atomic information, which could be a more convenient, faster, and effective supplement to experimental validation. In addition, we also use machine learning methods to analyze experimental data based on crystal structure, e.g., fast identification of crystal structures corresponding to XRD data, which could reveal pure phase and heterogeneous phase, etc. There are some useful results waiting to be compiled and published.
Reviewer 3 Report
Dear Authors! Thank you for your manuscript, submitted to "Materials". The presented investigation, devoted to the Machine Learning algorithm for the crystal structure and space group identification of perovskite compounds, is actual and well-timed, no doubt. Before the articles' publication, I would like to clearify some moments:
1. According to Eqs. (2), (3), ???, which is the bond valence between atom i and atom j, is calculated. What is the difference in the equations? What were the ??? values used in the subsequent LM technique (calculated from (2) or (3)?
2. From what database were the ??? values, the bond lengths between atom i and atom j, taken for calculations?
3. Is it possible to make the prediction calculations for the perovskites, not observed in Database? For example, for prognosis the existence of individual perovskite compound with preassigned composition?
Besides, some corrections would be provided into text:
1. Please, order the references correctly.
2. Line 40. Is it typo? (Cr, Mn, Sc - in A site)
3. Authors have noticed about the recent researches (Lines 276, 375). Please, add the references.
4. Please, add the correct references to Table 6.
5. It is wishable to delete the Ref [1] from Abstract.
Author Response
Dear Editors and Reviewers:
Thank you for your letter and for the reviewers’ comments concerning our manuscript entitled “Study on the automatic identification of ABX3 perovskite crystal structure based on the bond-valence vector sum”(ID: materials-2124174). Those comments are all valuable and very helpful for revising and improving our paper, as well as the important guiding significance to our research. We have studied the comments carefully and have made corrections which we hope meet with approval. Revised portions are marked in red on the paper. The main corrections in the paper and the responses to the reviewer’s comments are as flowing:
Responds to the reviewer’s comments:
Point 1: According to Eqs. (2), (3), Sij, which is the bond valence between atom i and atom j, is calculated. What is the difference in the equations? What were the Sij values used in the subsequent LM technique (calculated from (2) or (3)?
Response 1: Thank you very much for your questions on equations (2) and (3). Equations (2) and (3) are empirical formulas for two different relationships between bond valence Sij and bond length Rij summarized from a large amount of known crystal structure data. The relationship between valence strength and bond length was proposed by Donnay [1] and others in the 1970s and related through the form of inverse power function, and later the exponential relationship equation was proposed by Brown [2] and other scholars in Canada. The calculation used in this study is that of equation (3). We have removed the original equation (2) in the revised version(In line 121).
[1] Altermatt D, Brown I D. The automatic searching for chemical bonds in inorganic crystal structures[J]. Acta Crystallographica Section B Structural Science, 1985, 41(4): 240–244.
[2] Donnay G, Allmann R. How to recognize O2-, OH-, and H2O in crystal structures determined by x-rays[J]. American Mineralogist, 1970, 55(5–6): 1003–1015.
Point 2: From what database were the Rij values, the bond lengths between atom i and atom j, taken for calculations?
Response 2: Rij can be determined from the Inorganic Crystal Structure Database, Cambridge Structure Database.
Point 3: Is it possible to make the prediction calculations for the perovskites, not observed in Database? For example, for prognosis the existence of individual perovskite compound with preassigned composition?
Response 3: It is possible to predict unknown perovskites theoretically. Machine learning or deep learning method uses data (theoretical data, experimental data) to train models. The bigger amount of data used, the more stable and reliable the prediction model is constructed. In essence, machine learning or deep learning method uses nonlinear statistical methods to deduce the relationship between input parameters and output results, which is unknown and hard to describe by liner formula or model. If the prediction model is built stably and reliably, it is out of the question to prognosis the existence of individual perovskite compounds with preassigned composition.
Besides, some corrections would be provided into text:
- Please, order the references correctly.
- Line 40. Is it typo? (Cr, Mn, Sc - in A site)
- Authors have noticed about the recent researches (Lines 276, 375). Please, add the references.
- Please, add the correct references to Table 6.
- It is wishable to delete the Ref [1] from Abstract.
Response: Thank you for your reminder, we have rearranged the references in the revised manuscript and removed the typos in the original 40 lines, and updated the references by adding references [33,34] in line 280 of the revised manuscript, and adding references [18], [33], [34] in Table VI(In line 289), and removing the original reference [1].
- Park, W.B.; Chung, J.; Jung, J.; Sohn, K.; Singh, S.P.; Pyo, M.; Shin, N.; Sohn, K.S. Classification of crystal structure using a convolutional neural network. IUCrJ 2017, 4, 486-494 .
- Liang, H.; Stanev, V.; Kusne, A.G.; Takeuchi, I. CRYSPNet: Crystal structure predictions via neural networks. Physical Review Materials 2020, 4, 123802.
- Li, Y.; Dong, R.; Yang, W. Composition based crystal materials symmetry prediction using machine learning with enhanced descriptors. Computational Materials Science, 2021, 198, 110686.
Reviewer 4 Report
The comments can be found in the attached file.

Author Response
Dear Editors and Reviewers:
Thank you for your letter and for the reviewers’ comments concerning our manuscript entitled “Study on the automatic identification of ABX3 perovskite crystal structure based on the bond-valence vector sum”(ID: materials-2124174). Those comments are all valuable and very helpful for revising and improving our paper, as well as the important guiding significance to our research. We have studied the comments carefully and have made corrections which we hope meet with approval. Revised portions are marked in red on the paper. The main corrections in the paper and the responses to the reviewer’s comments are as flowing:
Responds to the reviewer’s comments:
Point 1: Please explore that for what reason, models with too few features are not appropriately trained in this approach?
Response 1: Thank you for your question. During the training process of a machine learning model, if the number of feature descriptors is too small, it will cause underfitting (high bias), and on the contrary, it will cause overfitting (high variance), so it is important to choose the right number of feature descriptors.
Point 2: The novelty and new strategy of the proposed approach should be more remarked.
Response 2: Thank you for your suggestions. We add our approach and strategy again in lines 282 to 288 of the revised version
Point 3: Among the offered models, which one is finalized?
Response 3: Thank you for your question. We did pre-training with multiple machine learning models and compared the performance, and finally found that Random Forest (RF) performed the best and was more sensitive to our data in classification and regression problems, and finally selected the Random Forest model.
Point 4: English language of the text should be revised. There are some grammatical mistakes in the text. Please check the text by a Native-English speaker.
Response 4: Thanks for your suggestion, we will take a native English speaker to check the grammatical mistakes.
Point 5: Conclusion part of the study should better present the achievements of the current study, which is a little unclear.
Response 5:Thank you for your suggestions on the conclusion section. We have taken your suggestions and rewritten the conclusion section(in line 375 to line 401).
Point 6: Why investigating the key feature factors that most influence the goal features is essential?
Response 6: Thank you very much for your question. In fact, machine learning or deep learning is the process to build a nonlinear model by nonlinear statistical methods essentially, and the feature variables serve as input parameters. The optimization of the input parameters plays a crucial role in the validity, reliability, and complexity of the model. It can also be seen in our study that when the input feature variables are reduced to 10, the recognition accuracy is improved continuously and the computation time is shortened. That reflects the significance of this study.
Point 7: In this regard, the following studies https://doi.org/10.1177/10775463211056758, https://doi.org/10.1016/j.compstruct.2022.115688 can be investigated.
Response 7: Thank you for your valuable suggestions, we have done research on the relevant literature.